# The Expression of Signaling Genes in Breast Cancer Cells

**DOI:** 10.3390/biology11040555

**Published:** 2022-04-03

**Authors:** Jolanta Rzymowska, Andrzej Wilkołaski, Lidia Szatkowska, Ludmiła Grzybowska

**Affiliations:** 1Department of Biology and Genetics, Medical University of Lublin, Chodźki 4A, 20-093 Lublin, Poland; 2Cardiology Clinic, Independent Public Clinical Hospital Nr 4, Chodźki 8, 20-093 Lublin, Poland; awilko@mp.pl; 3Department of Cardiology, 4th Military Clinical Hospital with Polyclinic SP ZOZ Wrocław, ul. Rudolfa Weigla 5, 50-981 Wrocław, Poland; lidia.szatkowska@op.pl; 4Department of Radiotherapy, Medical University of Lublin, Chodźki 7, 20-093 Lublin, Poland; ludmila.grzybowska-szatkowska@umlub.pl

**Keywords:** breast cancer, paclitaxel, signaling genes

## Abstract

**Simple Summary:**

The aim of the study was to investigate the effect of a drug for cancer—paclitaxel—on the expression of genes encoding the signaling factors in breast cancer cells outside organisms. The tested cells were harvested from the mammary glands of 36 women with breast cancer. The microarray technology —the carrier with applied DNA samples—was employed for the identification of gene expression. A significant effect of paclitaxel on the genome of breast cancer cells was confirmed. Paclitaxel changed the functions of cancer cell by increasing the expression of the genes encoding signaling proteins. This is the molecule of intercellular communication. The analysis of the results suggests that this cytostatic agent produces a beneficial therapeutic effect at a lower dose (60 ng/mL). In contrast, a high dose of paclitaxel (300 ng/mL) was associated with higher cytotoxicity and this had a negative effect on the tested tumor cells.

**Abstract:**

The aim of the study was to investigate the effect of paclitaxel on the expression of genes encoding signaling factors in breast cancer cells in in vitro conditions after incubation with the said chemotherapeutic. The tested cells were harvested from the mammary glands of 36 patients with early breast cancer. The microarray technology was employed for the identification of gene expression. For this purpose, mRNA isolated from tumor cells was used. A significant effect of paclitaxel on the genome of breast cancer cells was confirmed. Paclitaxel changed the functions of cancer cells by increasing the expression of most genes encoding signaling proteins and receptors. The analysis of the results suggested that this cytostatic agent produces a beneficial therapeutic effect at a lower dose (60 ng/mL). In contrast, a high dose of paclitaxel (300 ng/mL) was associated with a high cytotoxicity.

## 1. Introduction

Breast cancer is a malignant tumor that derives mostly from the epithelium of the terminal ducts. It is the most common malignancy in women worldwide. Genetic, hormonal, and environmental factors play a role in its pathogenesis [1].

The transmembrane receptor Wnt and its co-receptor LRP-5/6 protein (lipoprotein receptor-related proteins 5 and 6) inhibit the degradation of beta-catenin, which accumulates in the cytoplasm and then moves to the nucleus [2]. Beta-catenin binds the transcription factors TCF (T-cell factor)/LEF (lymphoid enhancer factor) and stimulates gene expression, for example *c-myc*, *c-jun*, and *fra-1* encoding cyclin D1. 

The coupling of a ligand with a Wnt receptor leads to the activation of one canonical β-catenin pathway and two non-canonical pathways. The intracellular signaling cascade is different for each pathway, with the only common element being the binding of the ligand with the Wnt FZD receptor [3]. The Wnt-1 signaling pathway plays a key role in the functioning of cells, but its exact role in the development of breast cancer has not yet been elucidated. It has been shown that Wnt-1 expression in breast cancer cells is significantly higher than in healthy cells, and breast cancer growth is enabled by their rapid proliferation. Wnt-5a overexpression increases the invasiveness and metastatic potential of cancer cells and is closely correlated with cancer progression [4,5]. 

Beta-catenin is a protein whose expression increases during molecular carcinogenesis. In normal conditions, it is produced by linking cell adhesion proteins (called cellular adhesion molecules, CAMs) with the cytoskeleton. It also regulates the expression of other genes that control cell proliferation and play a role in the intracellular Wnt/Wingless signaling pathway. Its main regulator protein is actin which forms a complex of APC and glycogen synthase kinase [6,7]. This complex binds beta-catenin to regulate the size of the free fraction and its intracellular distribution. In the event of the APC activity loss, non-beta-catenin accumulates in the nucleus, acting as a co-activator of LEF/TCF transcription factors (lymphoid enhancer factor /T-cell factor) [7]. This triggers the transcription of oncogenes, such as *c-myc*. The WNT activation pathway inhibits APC activity, thereby blocking beta-catenin phosphorylation, which increases its intracellular concentration [4]. 

RAB32 a part of the binding GTP with the oncogenic *Ras* family. The *Ras* gene encodes the membrane-associated proteins of the G protein family, which transmit information from multiple cell surface receptors to the enzyme mediating the formation of secondary messengers. The activity of the G protein in connection with GTP is controlled by autoregulation. The *Ras* oncogene with a point mutation shows no GTPase activity. Although most breast cancers (95%) have no *Ras* mutations, there is a relationship between the over activity of this gene and the development of carcinogenesis. In breast cancer cells, *Ras* expression was reflected in high mitogen-activated protein kinase (MAP kinase) activity as well as EGFR (Epidermal Growth Factor Receptor) and erbB-2 overexpression [8,9,10]. 

The family of G protein signaling regulatory proteins (RGS) controls the activity of G proteins by accelerating the GTPase activity of the alpha subunits of proteins. Most likely, RGS proteins control the functioning of cancer cells, including tumor progression [11].

BCAR 3 (Breast Cancer Antiestrogen Resistance-3) is found in fibroblasts and is expressed in breast cancer cells not expressing the estrogen receptor (ER). It is a gene that plays a role in the migration and invasion of cancer cells. BCAR3 mRNA expression is undetectable in normal breast cancer tissue [12].

Antymitotic taxanes have been used in the treatment of breast cancer for many years. Paclitaxel and docetaxel are widely used in the treatment of cancer. These drugs are prepared from semisynthetic needles of the European yew tree (Taxus baccata). The difference in the structure of these substances involves two side groups of carbon atoms 10 and 13. The main known mechanism of the anticancer action of taxanes is the effect on microtubules by enhancing tubulin polymerization and inhibiting depolymerization. There are also reports that taxanes impact on microtubule-associated proteins (MAPs), cell cycle proteins, and apoptotic pathway proteins [13,14]. 

The main target of taxanes is the microtubule cytoskeleton. The interaction between disrupted microtubules and taxanes prevents the correct separation of sister chromatids in homologous chromosomes during cell division, which leads to cell division arrest [13,14]. The action of taxanes, as the action of other mitotic inhibitors, is phase-specific and relates to the S, G2, and M phases. Cell cycle arrest at the G2/M phase launches a number of mechanisms responsible for apoptosis.

Paclitaxel has a different effect on the signaling pathways depending on its concentration in the cell. Low concentrations of taxol inhibit progression through the dysfunction of a microtubule that cannot polymerize properly. It was also found that low drug concentrations induce p53 and p21WAf independent of the RAF-1 pathway. However, at higher concentrations, cell death occurs as a result of the blocking of the end stage of mitosis which is dependent on the RAF-1 pathway. Paclitaxel-mediated cell death may result from Raf-1 dependent and independent pathways [15].

Paclitaxel is characterized by high effectiveness both in early breast cancer and in metastatic breast cancer. Primary or acquired drug resistance of tumor cells to taxanes is a significant clinical problem. Among the known mechanisms of the resistance of breast cancer cells to paclitaxel, the most important are the active removal of the drug from the cell. This is related to the increased activity of ABC family membrane transporters which are closely related to the function of the cellular signaling system, changes within the molecular targets of this cytostatic, or apoptosis.

Thanks to significant advances in molecular genetics and the emergence of the microarray technique, it is now possible to assess the profiles of the DNA, mRNA, and proteins in tumor cells and body fluids. This technology can be used to assess gene expression, i.e., to quantify a particular gene product—mRNA. The microarray technique allows one to determine the expression of tens of thousands of genes in a single test [16,17]. 

In the case of breast cancer, microarray studies have led to the emergence of genetic profiles which have enabled the following: 

### 1.1. The Distinction between Two Main Types of Breast Cancer and its Subtypes with Different Prognoses


characteristic profile of glandular cells forming the inner layer of normal ducts and lobules (two subtypes A and B) [1];profile showing no expression of genes characteristic of the said profile (three subtypes) [18,19].


### 1.2. Predicting Distant Metastases and Survival (MammaPrint 70 Genes), and the Assessment of the Risk of Recurrence (OncotypeDx, 21 Genes)

Cancer cells have a different metabolism than normal cells. 

In the process of neoplasms, the differentiation, survival, and proliferation of cells are disturbed. Resistance to applied chemotherapy, (including paclitaxel), apart from active drug removal from the cell, is also caused by disturbances in the expression of apoptotic regulatory proteins, loss of resistance to proapoptotic factors, inhibition of apoptosis, and activation of autophagy. The studied genes, signaling pathways, and protein receptors participate directly or indirectly in the processes responsible for both proliferation and differentiation as well as cell death.

Thanks to the use of genomic arrays, we can test mutations and gene expressions in cancer cells. The investigation of breast cancer using genetic techniques has revealed the existence of groups of cancers with significantly different profiles. The use of DNA microarrays has opened new opportunities for understanding the biology of cancer and has given hope to improve treatment outcomes.

The aim of this study was to investigate the effect of paclitaxel on the expression of genes encoding signaling factors in breast cancer cells by:
the identification of a group of gene-encoding factors involved in signaling in breast cancer cells which showed differences in expression after administration of the chemotherapeutic agent;the determination of statistical significance and correlation values for the expression of individual genes and their importance for the prediction of cancer progression and prognosis.


## 2. Materials and Methods

### 2.1. Cell Cultures 

The study involved cells obtained from the breast tissue of 36 patients diagnosed with early stage I and II breast cancer (T1-2N0M0,T—tumor, N—nodes, M—metastasis) according to the WHO (World Human Organization), histopathological results of no special type (NST), and G1 and G2 ductal carcinoma (G—grade) according to the previous histopathological classification. The patients presented with triple-negative breast cancer (estrogen, progesterone, and Her2 receptors were all negative). Patients’ ages ranged from 46 to 76 years, with a mean age of 56. The patients had not received any chemotherapy or hormonotherapy. They were all perimenopausal or postmenopausal and were women. The recruitment hospital was the Oncology Center of Lublin, and the recruitment period was from 2010 to 2012.

The research material was isolated from the tumor itself, without margin. It was a material verified by a pathologist, the second part of which was used for histopathological examinations. Elimination of cells outside the tumor (blood vessel cells, fibroblasts, cells of the immune system, necrotic cells, apoptotic cells) was completed during the first 3 days of culture thanks to the use of sterile 0.2 µm membrane filters and adherent culture vessels. The karyotype was determined after the first passage to rule out genomic instability. 

Samples were taken during surgery and stored at 4 °C. All homogenization and RNA isolation procedures were carried out at 4 °C. The material was subjected to mechanical homogenization and enzyme disintegration (0.1% trypsin, EDTA SIGMA, Poland). Cell cultures were established in disposable sterile plastic vessels in RPMI medium (Sigma, Poland) with 10% fetal bovine serum (FBS, Sigma, Poland) as well as penicillin and streptomycin. 

Isolated cells were cultured in an incubator (37 °C, 5% CO_2_, 90% air humidity). Paclitaxel (Bristol Myers Squibb) was added at concentrations of 60 and 300 ng/mL and incubated for 72 h in cultures with a density of 10,000 cells/mL. The concentrations were calculated on the basis of dose per square meter of body surface area in relation to the surface of the culture vessel (25 cm^2^). The concentrations corresponded to the doses of the drug used in mono- and polytherapy for breast cancer, taking into account the number of paclitaxel chemotherapy cycles (approximately six). Control culture suspensions contained breast cancer cells that were incubated without the cytostatic in a medium containing 5% dimethyl sulfoxide (DMSO). 

### 2.2. Examined Genes

Fifteen genes encoding signaling proteins, which may have had diagnostic applications, were selected. The spelling for genes and proteins was adopted in accordance with the instructions, where the names of human genes are spelt with capital letters in italics. For names of genes other than human genes, capital letters are used in Table 1, while the names of proteins are spelt in non-italicized capital letters.

### 2.3. Total RNA isolation

Cells isolated from tumors of the mammary glands (the primary, established cultures) after incubation with cytostatic and without it were homogenized in TRI solution (SIGMA). Separation at 4 °C was conducted, and the obtained homogenate was centrifuged for 10 min at 12,000× *g*. The aqueous fraction was transferred to a sterile tube with 0.5 mL of isopropanol. After a 10-min incubation at room temperature, the cells were centrifuged again at 12,000× *g* at 4 °C. A total of 1 mL of 75% ethanol was added to the resulting RNA-containing precipitate and centrifuged at 8000× *g* at 4 °C. Total RNA (RNA T) was checked for protein contaminants and DNA additives using a spectrophotometer (Eppendorf) measuring light absorbance by the aqueous RNA solution in the T 1:100 dilution at wavelengths of 260 and 280 nm. For further analysis, an RNA sample was collected for which the absorbance ratio was in the range 1.6–1.9. The resulting RNA T contained a small amount of mRNA, and mostly rRNA and tRNA. 

### 2.4. Reverse Transcription Polymerase Chain Reaction (RTPCR)

Complementary DNA (cDNA) was obtained in reverse transcription polymerase chain reaction (RTPCR) using a standard kit (Sigma) and was subsequently subjected to hybridization (Set Panorama Human Cancer cDNA labelling kit and hybridization; CDLBL-HCNGENOSIS SIGMA). The set of reference RNA was derived from the reference *Escherichia coli* gene (RNA Panorama *E. coli* B-1444 RNA) that enabled the subsequent calibration of the activity of other genes present on the surface of the array relative to the expression of this gene. Samples containing 2 µg of total RNA, 4 µL of standard primers, and 2 µL of the reference *Escherichia coli* RNA (control RNA–HAV RNA in vitro transcript) were completed with water to a 14.5 µL. They were then heated for 2 min at 90 °C and cooled to 42 °C for another 20 min. The reaction mixture consisted of nucleotides dATP, dGTP, dTTP, and dCTP labelled with 40 µCi ^32^P in a volume of 4 µL, 4 µL reverse transcriptase, and 6.5 µL of a reverse transcriptase buffer. The ingredients were incubated at 42 °C for 150 min. Afterwards, the reverse transcription method was conducted to obtain cDNA. (Transcriptor One-Step RT-PCR Kit Sigma). This kit included a protector RNase inhibitor and a control primer mix HAV RNA (forward and reverse primers) [20].

The purification process was conducted in a Sephadex G-25 by introducing the cDNA aqueous solution. The material in the column was centrifuged for 4 min at 11,000× *g*.

### 2.5. cDNA Array Hybridization

The cDNA purified in this way was subjected to array hybridization. The SIGMA-GENOSIS matrix, containing gene sequences encoding the signaling factors, was used for testing. 

Hybridization was carried out at 65 °C for 13 h. The array matrix was washed with 100 mL of a washing buffer and incubated in a hybridization oven at 65 °C for 60 min, then centrifuged at the speed of 6 revolutions per min. This step was repeated twice.

After washing and removing the cDNA, free substrate fragments were dried on sterile filter paper for 2 min and placed in a cassette capturing gamma radiation (radiation imaging screen, Bio-Rad). The measurement of array activity took about 24 h. Gamma radiation from spots corresponding to the genes on the array caused chemical reactions of the radiation-sensitive medium contained in the cartridge. After irradiation, the medium was scanned with a Molecular Image FX scanner (Bio-Rad) with a resolution of 50 microns. This way the image of gene expression was shown on the array. The expression of the array in individual spots was the normalized gene expression of the reference *Escherichia coli.* Each of the fields corresponded to the area of a single gene on the array. The fields were segmented in larger groups (spots). A single spot on the array had 16 fields, which corresponded to the expression of 8 genes, while the remaining 8 were blank.

### 2.6. Analysis of the Array Gene Expression Matrix

The analysis of the array gene expression was performed in the Quantity One program, version 4.2.1. The expression of genes in different spots presented an average number of pixels in the area specific for a given gene. Each of the two adjacent fields in one row was a signal region corresponding to the spot of a given gene. The values of these numbers directly represented the intensity of the signal received from the bottom of the radiation-sensitive cassette and corresponded to the expression level of particular genes. Gene expression analysis was repeated three times.

The analysis of the results was based on a comparison of the gene expression in breast cancer cell cultures incubated with two doses of paclitaxel and expression of these genes in cultures incubated without the drug.

The study was completed in three independent replicates involving three control cultures and three cultures with lower and higher doses of paclitaxel.

### 2.7. Statistical Analysis

The statistical analysis was performed using Statistica 10.0 and Microsoft Excel 2015. We computed the arithmetic average and standard deviations for individual groups of genes. In order to assess the significance of the differences between individual groups of genes, we used the Student’s *t* test. For multiple comparisons, one-way analysis of variance (ANOVA) followed by Spearman’s rank post-hoc test was applied. Statistical significance was considered as *p* < 0.05. The rank correlation takes values in the range [−1, +1].

Functional validation analysis of candidate genes and network analysis of key nodes were performed in the program RPANTHER™ GO slim (version 17.0, based on GO release 16 November 2021, released 22 February 2022) [21].

## 3. Results

Except for the *WNT3* and *CHN2* genes, all examined signaling proteins in Table 1 showed increased expression in breast cancer cells incubated with both the low dose of paclitaxel (60 ng/mL) and the high dose of this drug (300 ng/mL) as compared to the control cells. In the case of *WNT3* and *CHN2,* the lowest expression was observed for the dose of 300 ng/mL (Figure 1 and Figure 2H). The *CHN2* expression level was the same in the control cells and in the samples incubated with 60 ng/mL of paclitaxel (Figure 2H).

The highest expression after incubation with paclitaxel at the dose of 300 ng/mL was noted for the following oncogenes: *WNT10, RAB32, SR-BP1, BCAR3, LAMC1,* and *CLDN5* (Figure 3A–F).In the case of *BCAR3,* the expression was similar for the two doses of paclitaxel (60 ng/mL vs. 300 ng/mL) (Figure 3D). Moreover, slight differences were observed between the tested doses in the expression levels of *WNT10* and *RAB32* (Figure 2A,D).

The highest gene expression was observed for the dose of 60 ng/mL, and the expressed genes were: *CTNB1*, *RGS2*, *RGS16, LAD1, LBP, TAP2, PKP3,* and *CHN2* (Figure 2A–G). *CHN2* expression was high, but still at the same level as the control (Figure 2H). Slight dose-dependent differences in the expression levels were noted for *LAD1 and TAP2 (*Figure 2D,F) as well as *WNT10 and BCAR3* (Figure 3A,D).

Analysis of the correlations of gene expression showed that they were statistically significant (Table 2).

Analysis by Spearman’s test showed that the increase in WNT 3 gene expression was accompanied by an increase in WNT10 and TREM1 expression at both the 300 mg and 60 mg doses. There were also positive correlations between RAB32 and RGS2, PKP3 and RGS16, and TREM 1 and PKP3. In the remaining cases, no such correlation was demonstrated (negative correlation).

The results of functional validation analysis of candidate genes and network analysis of key nodes were presented in Figure 4, Figure 5 and Figure 6. Each of the candidate genes is involved in biological processes or in the molecular function of cells. Five genes (WNT10B, CTNNB1, WNT3, RGS16, RGS2, LAMC1) also play a role in the signaling pathway—Figure 5.

## 4. Discussion

Healthy and neoplastic tissues have different mechanisms of regulation not only of the expression of multi-drug resistance genes, e.g., MDR1, MRP1, and ABCG2, but also genes whose protein products indirectly regulate the activity of ABC transporters, including from the WNT/β-catenin pathways. The CHN2 B2 gene product—chimerin—is responsible for the progression of breast cancer. Cancer cells also overexpress PKP3 which is a WNT antagonist. High levels of PKPs are associated with a shorter and more specific survival periods. B-catenin activates the genes of the transcriptional complex, which leads to tumor progression [22]. The Wnt signaling pathway plays an important role in carcinogenesis. Disturbances in the functioning of this pathway, such as excessive beta-catenin accumulation, are frequently observed in breast cancer cells. It has been noted that Wnt affects breast cancer development to a significant extent, and its increased activity is associated with a poor prognosis. In contrast, a pathway blockage leads to the inhibition of cell proliferation and reduction in the cell oncogenic potential. This pathway plays a key role in inhibiting the oncogenic potential of breast cancer [23,24].

Although Wnt activation in breast cancer cells has been reported, it is rarely caused by mutations in genes encoding proteins of this pathway. Up regulation of the WNT LRP6 signaling pathway receptor has been noted in breast cancer cells, and its likely impact on the development of neoplastic processes has been demonstrated. In breast cancer cells, there is overexpression of *WNT3A*, *WNT4*, *WNT6*, *WNT8B*, *WNT9A* and *WNT10B* [25,26,27]. Beta-catenin is a regulator protein of the *WNT* pathway. The expression of *WNT5A*, *WNT5B*, and *WNT16*, in turn, is generally reduced. It is worth noting that the WNT signaling pathway components undergoing increased expression in breast cancer cells belong to the elements of the canonical WNT/beta-catenin pathway. It is believed that auto- and paracrine mechanisms lead to the excessive activation of the WNT/beta-catenin canonical signaling pathway [28]. Elevated Wnt signaling is responsible for resistance to cancer therapy by maintaining the cancer stem cell population and enhancing DNA damage repair. Up-regulation of the WNT pathway after Paclitaxel administration has been observed, which may protect cancer cells from cell cycle arrest and apoptosis [29].

Studies suggest that the WNT signaling pathway may be a potential therapeutic target in inhibiting metastasis to the lungs and bones.

In the present study, *WNT3* expression was higher in the control cells than in the cells incubated with paclitaxel. In this case, the lowest expression occurred at a dose of 300 ng/mL. Both doses of paclitaxel resulted in a reduced *Wnt3* expression, but the higher dose caused a more significant decrease. Genes encoding the Wnt10 protein, however, exhibited increased activity under the effect of paclitaxel, at both doses. It is believed that overexpression of the *Wnt10* gene encoding the protein plays a key role in carcinogenesis through the β-catenin-associated activation pathway [30,31].

CTNNB, a protein associated with catenin (activator catenin), has an important role in intercellular adhesion and cell growth regulation. The expression of this protein is closely linked to oncogenesis. Mutations in the gene encoding this protein lead to the development of various cancers. Research has confirmed the role of beta-catenin in the development of breast cancer, and changes in its expression have been observed at early stages of carcinogenesis [32]. CTNNB is a key element of the canonical Wnt pathway. Research has shown a relationship between Wnt/beta-catenin and oncogenesis. Elevated levels of beta-catenin within the nucleus contribute substantially to the development of breast cancer [33]. In the present study, we demonstrated increased *CTNNB* expression at a reduced dose of paclitaxel.

Breast cancer growth is usually estrogen-dependent [1]. Anti-estrogens, such as tamoxifen, are widely used in breast cancer treatment. However, the development of this type of cancer can lead to its resistance to the effects of estrogen. This protein belongs to cytoplasmic receptors involved in intracellular signal transduction. Its activity leads to increased estrogen-independent breast cancer cell proliferation and resistance to tamoxifen. BCAR3 is functionally linked to another protein causing anti-estrogen resistance (p130Cas) [33,34]. It has been shown that there is a relationship between *BCAR3* expression and the regulation of breast cancer cell migration through a change in the location of BCAR1 in the cytoplasmic membrane. An increase in *BCAR3* activity has been observed in advanced breast cancer cells, indicating a relationship between intracellular signal transduction through BCAR3 and the intensity of metastasis. Recent studies have shown that *BCAR3* leads to the inhibition of breast cancer progression through blockage of the TGFß/Smad signaling pathway [34].

RAB, a GTP-binding protein, belongs to the family of oncogenic *Ras*. It belongs to the G protein family that transduces information from cell surface receptors to the enzyme mediating the formation of secondary messengers. Expression of RAB proteins influences the regulation of many cellular pathways. By affecting cell membranes, they affect signaling pathways. Their expression is disrupted in Alzheimer’s disease and cancer. RAB family genes are most likely overexpressed rather than mutated in tumors [35].

Although most breast cancers (95%) have no *RAS* mutations, there is a relationship between its overexpression and cancer development. *Ras* expression is reflected in the high activity of mitogen-activated protein kinase (MAP kinase) [8,9,36]. The expression of the oncogene for *Rab32* in our study was increased under the effect of paclitaxel.

Beta-CHN2 chimerin plays an important role in axonal growth and T-cell gene activation [37]. Beta-chimerin is a protein affecting cell proliferation and the migration of smooth muscle, and its expression is accompanied by the development of highly malignant breast cancer [38].

Another of the studied genes is LAMC1, the product of which is the LAMC1 transmembrane protein involved in cell adhesion and migration. It was shown that overexpression of lamin A/C caused the inhibition or stimulation of cell growth, colony formation, migration, and invasion [39]. Laminin is a major glycoprotein component of non-collagenous membranes. It is produced via numerous biological processes, such as adhesion, differentiation, migration, and intercellular signaling, as well as in tumor metastasis. Laminin consists of three different chains: alpha, beta, and gamma. Each of these is coded by a separate gene. The gamma chain oflaminin 1 is coded by *LAMC1.* Studies on breast cancer cells have demonstrated the effects of a “genome guardian” which cooperates with P53 inhibitor, i.e., miR-205 for LAMC1, and is responsible for the adhesion, proliferation, and migration of cancer cells. It is found in an inactive form associated with guanosine diphosphate (GDP) and in active GTP (guanosine-5’-triphosphate) [40,41].

The *SR-BP1* gene encoding the protein plays a role in the regulation of mature mRNA formation (splicing). Several variants of this protein have been found as a result of alternative splicing. The first step of the cascade leading to cancer is increased cell proliferation that reduces their differentiation and leads to the loss of physiological junctions. The claudin 5 encoding gene is a membrane protein involved in intercellular adhesion. It is a component of occluding junctions (also called tight junctions or zonula occludens), barriers for water and water-soluble nutrients.

Endothelial cells and some epithelial cells exhibit *CKDN5* expression [41,42]. Based on endothelial cells, it was found that abnormally functioning short circulating molecules can facilitate the spread of cancer. Presumably, this may affect breast cancer cell potential for hematogenous metastases. Increased expression of the gene encoding claudin 2 and 5 is characteristic of cancer. There is no relationship with overexpression of estrogen or progesterone receptors, or with cancer stage. No increase in the expression of the claudin-coding gene has been observed in skin lesions, which indicates that claudin may be a useful tool in the differential diagnosis of Paget’s disease and basal cell carcinoma [43,44]. On the other hand, no differences in the expression of claudin in Paget’s disease and breast cancer suggest the lack of connection between the functioning of these proteins and the degree of epithelial invasion. Due to an increase in the activity of claudin proteins in many cancerous processes, they have become a promising target for antitumor therapy. *CLDN5* overexpression has been shown particularly in endothelial cells that serve as potential targets in angiogenesis-inhibiting therapy. Claudin 5 plays an important role in the regulation of cell motility. This is believed to be associated with the regulation of the permeability of the blood–brain barrier [45,46,47,48,49].

A possible link has been revealed between claudin 5 and breast cancer metastases. Microarray and immunohistochemical studies have shown that the decreased expression of the claudin gene is associated with a worse prognosis and an increased tendency of cancer to metastasis to the regional lymph nodes [44,49].

The *RGS2* gene codes a protein that regulates intracellular signaling via G protein. RGS2 is an important factor in the cascade of signaling proteins initiated by the G protein. RGS2 inhibits signal transduction by increasing the activity of GTPase that affects the alpha subunit of the G protein. Due to conjugated signal transduction effector cells, RGS2 is also known as an angiotensin II receptor regulator. The molecular analysis of the isolated epithelial cells showed *RGS2* overexpression in breast cancer cells and the modulation of the signal transduced by the oxytocin receptor for the protein. The identification of a gene product and the pathways that regulate it appears to be an interesting therapeutic target. Studies in mice have shown that *RGS2* deletion leads to a significant inhibition of tumor processes by reducing the blood supply to the tumor as a result of angiogenesis inhibition [50,51] In studies on ovarian cancer, cells that showed low *RGS2* expression were characterized by considerably greater chemotherapy resistance. Recent studies have also suggested that there might be a relationship between *RGS2* overexpression and the presence of metastases. Differences in gene expression have occurred at various cytostatic doses. In the case of *RGS2,* the expression was the highest in tumor cells treated with 300 ng/mL of paclitaxel and, for the *RGS16,* the difference was the highest for 60 ng/mL.

The phosphatidylinositol pathway significantly contributes to the development of various cancers, including breast, lung, and prostate cancer. Breast cancer resistance to chemotherapy, including tyrosine kinase inhibitors, is associated with the constant activation of the phosphatidylinositol pathway, which may cause increased expression of epidermal growth factor receptors (HER2 and HER3). In our studies, *TAP2* expression was higher at the lower dose than the higher dose. At higher doses, the multidrug resistance process might be interrupted, which may be elucidated with a decreased *TAP2* expression. Recent studies have shown that the RGS family inhibits the activity of this pathway. In light of the previous observations, reduced *RGS16* expression is observed in the majority of breast cancers with mutations, which supports the existence of an association of phosphatidylinositol pathway activity regulation with *RGS16* expression and breast cancer cell growth [51]. *RGS16* overexpression in breast cancer cells inhibits the cell prolife ration dependent on epidermal growth factor (EGF), and decreased *RGS16* expression was associated with a reduction in the cell proliferation rate and an increased resistance to tyrosine kinase inhibitors. The implementation of tyrosine kinase inhibitors in therapy also causes decreased *RGS16* expression in breast cancer cells. It seems that *RGS16* may play an important role in controlling breast cancer cell growth through its influence on the key pathway leading to cell death [50,51].

It is believed that beta-catenin activator *CTNNB1* mutations, which induce disturbances in Wnt/beta-catenin functioning, lead to increased accumulation of beta-catenin in the nucleus and have a key function in the development of breast cancer [3,4]. Such changes are observed frequently in breast cancer cells. Beta-catenin plays an important role in the carcinogenesis of ductal carcinomas, and changes in its expression are observed in in situ breast cancer, suggesting that these expression changes are connected with even early stage carcinogenesis. Disorders of Wnt/beta-catenin functioning may also be associated with the occurrence of metastases. The identification of a mutation in *CTNNB1* may lead to the development of new treatments for breast cancer patients. There is a relationship between the beta-catenin expression in the nucleus and a higher incidence of metastases and lower survival in women with breast cancer. The activation of the Wnt/beta-catenin pathway is associated with a worse prognosis, although Geyer et al. [52] did not notice any correlation between *CTNNB1* mutations and prognosis.

The Wnt/beta-catenin pathway plays a key role in breast cancer progression [24]. For this reason, the effect of down-regulation in breast cancer cells was evaluated in mice. A significant reduction in beta-catenin activity was associated with reduced tumor proliferation. Other studies on reduced beta-catenin expression in breast cancer cells have shown that these cells tend to produce significantly smaller tumors and that their growth is considerably slower compared to the control group with normal beta-catenin activity. This increased tumor sensitivity to both doxorubicin and cisplatin chemotherapy. In addition, earlier reports indicating that the canonical Wnt pathway activation is associated with more rapid development of cancer and poor prognosis were confirmed [53,54,55].

In the present study, we demonstrated increased *CTNNB1* expression after cell incubation with 60 ng/mL of paclitaxel. We also showed a higher expression of most genes encoding signaling proteins in cells treated with paclitaxel. The data analysis revealed the increased expression of genes encoding receptor proteins in cell membranes [56].

Moreover, the highest expression of the LBP gene was observed at the 60 ng dose. The product of *LBP*, the lipopolysaccharide binding protein (LBP), is a serum protein that binds and transports LPS (lipopolysaccharide ) It is associated with a worse prognosis in colorectal and renal carcinoma [57,58].

Our results have shown different effects of the two doses of paclitaxel (60 and 300 ng/mL) on the expression of signaling genes in breast cancer cells in in vitro settings. The changes in the genes of the test subjects appeared with two different doses of paclitaxel. This may have been due to intracellular interactions, but it was directly or indirectly due to the effects of the two different doses of the drug used. The mechanism of these changes is difficult to define unequivocally. The analysis of the results suggests that this cytostatic agent produces a beneficial therapeutic effect at a lower dose (60 ng/mL). In contrast, a high dose of paclitaxel (300 ng/mL) was associated with a high cytotoxicity. The obtained results may be useful in the molecular work-up in patients with a breast tumor. The expression of signaling genes can be of prognostic significance and may be used as a predictor of breast cancer treatment.

## 5. Conclusions

In the process of neoplasms, the differentiation, survival, and proliferation of cells are disturbed. Resistance to applied chemotherapy, (including paclitaxel), apart from active drug removal from the cell, is also caused by disturbances in the expression of apoptotic regulatory proteins, loss of resistance to prapoptotic factors, inhibition of apoptosis, and activation of autophagy. The analysis of the results suggests that paclitaxel showed a more favorable therapeutic effect at the lower dose (60 ng/mL) than at the high dose (300 ng/mL), which was associated with a high cytotoxicity. The studied proteins, signaling pathways, and receptors genes participate directly or indirectly in the processes responsible for both proliferation and differentiation as well as cell death.

## Figures and Tables

**Figure 1 biology-11-00555-f001:**
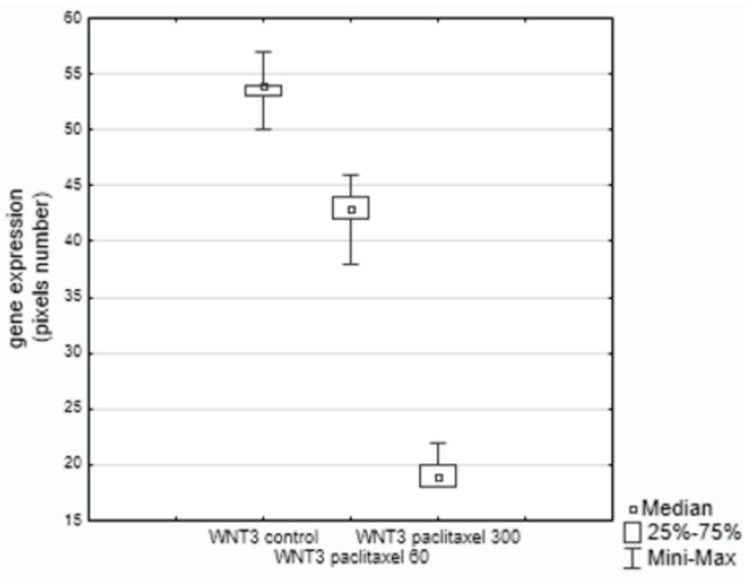
A significant difference between the control culture (*WNT3* control) and the cultures with paclitaxel (*WNT3* paclitaxel), and between cultures with 60 and 300 ng of paclitaxel; *p* = 0.000001.

**Figure 2 biology-11-00555-f002:**
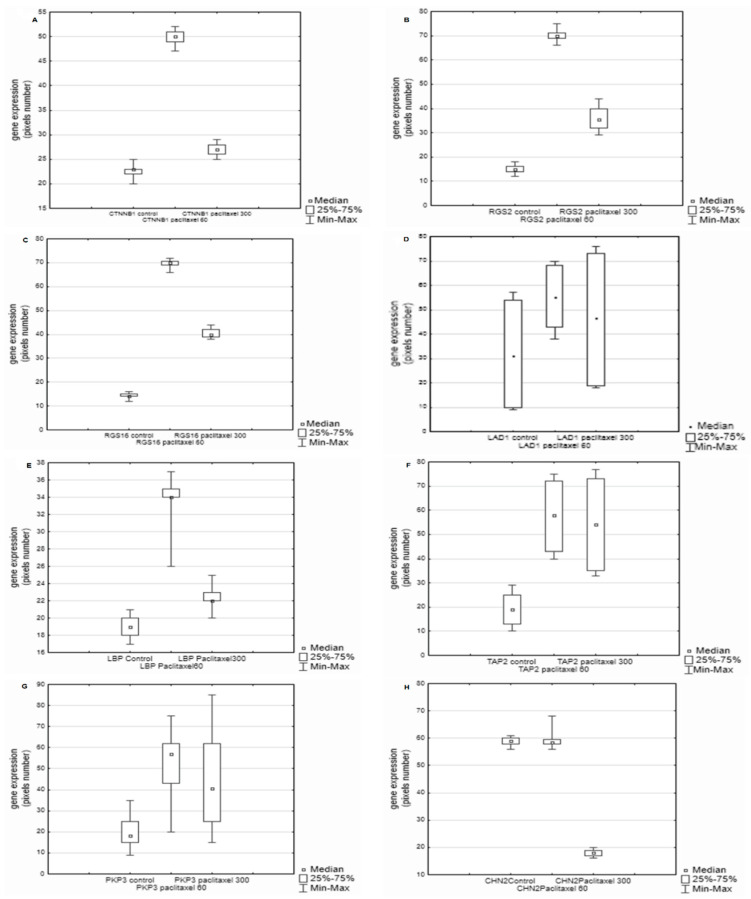
Differences in the expression of exanimated genes between the control cultures and the cultures with paclitaxel, and between the cultures with two examined doses of paclitaxel. (**A**) A significant difference between the (*CTNNB1*) control culture and the cultures with paclitaxel, and between cultures with two examined doses of paclitaxel; *p* = 0.000001; (**B**) A difference between the control culture (*RGS2*) and the cultures with paclitaxel, and between cultures with different doses of the cytostatic; *p* = 0.000001; (**C**). A significant difference between the control culture (*RGS16*) and the cultures with the cytostatic, and between cultures with different doses of paclitaxel; *p* = 0.000001; (**D**). A difference between the control culture (*LAD1*) and the cultures with paclitaxel *p* = 0.000001; (**E**). A significant difference between the control culture (*LBP*) and the cultures with paclitaxel, and between cultures with two doses of paclitaxel; *p* = 0.000001; (**F**). A difference between the control culture (*TAP2*) and the cultures with paclitaxel; *p* = 0.000001. There was no difference between the cultures with two doses of the cytostatic; (**G**). A significant difference between the control culture (*PKP3*) and the cultures with paclitaxel. There was no significant difference between the cultures with the examined doses of paclitaxel; *p* = 0.000001; (**H**). A difference between the control culture (CHN2) and the cultures with a higher dose of paclitaxel. There was no significant difference between the control culture with the lower dose of paclitaxel; *p* = 0.000001.

**Figure 3 biology-11-00555-f003:**
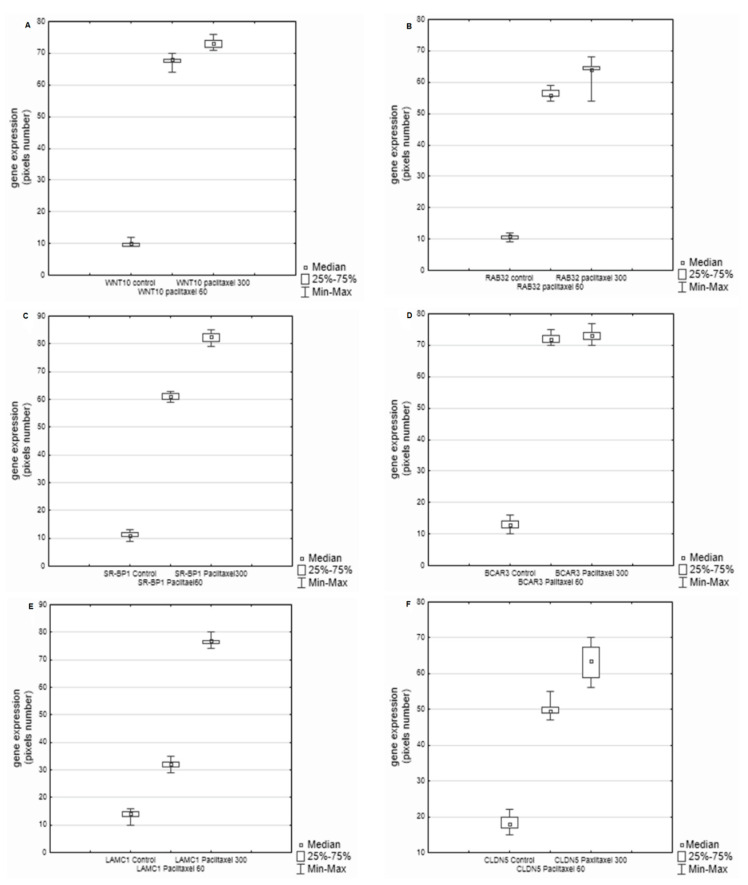
Differences in the expression of exanimated genes between the control cultures and the cultures with paclitaxel, and between the cultures with two examined doses of paclitaxel (60 and 300 ng ); (**A**). A significant difference between the control culture (*WNT10B* control) and the cultures with paclitaxel (*WNT10B* paclitaxel) and between cultures with different doses of paclitaxel; *p* = 0.000001. (**B**) A difference between the control culture (*RAB32*) and the cultures with paclitaxel, and between cultures with different doses of paclitaxel; *p* = 0.000001. (**C**). A significant difference between the control culture (*SR-BP1*) and the cultures with paclitaxel, and between the cultures with different doses of the drug; *p* = 0.000001. (**D**). ANOVA showed a difference between the control culture (BCAR) and the cultures with paclitaxel; *p* = 0.000001. There was no significant difference between the cultures with the examined doses of paclitaxel. (**E**). A difference between the control culture (LAMC) and the cultures with two doses of paclitaxel; *p* = 0.000001. (**F**). ANOVA showed a significant difference between the control culture (*CLDN5*) and the cultures with paclitaxel, and between the cultures with different doses of the drug; *p* = 0.000001.

**Figure 4 biology-11-00555-f004:**
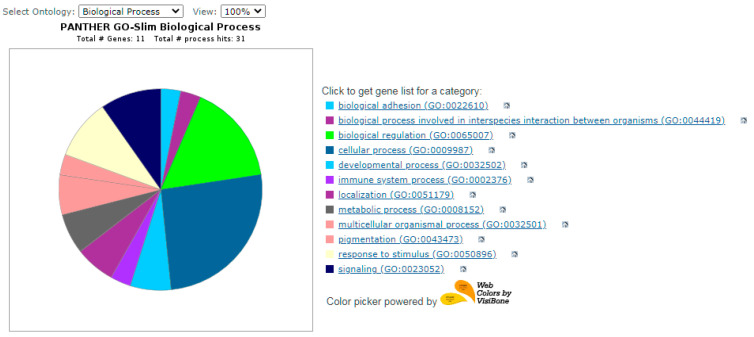
Tested genes and the biological process. 

biological adhesion (GO:0022610) 

- CLDN5, PKP3,

biological process involved in interspecies interaction between organisms (GO:0044419) 

- LBP, 

biological regulation (GO:0065007) 

 -LBP, WNT10B, SREBF1, CHN2, BCAR3, CTNNB1, WNT3, 

cellular process (GO:0009987) 

-LBP, WNT10B, SREBF1, RAB32, CLDN5, PKP3, BCAR3, TAP2, CTNNB1, WNT3, 

developmental process (GO:0032502) 

-WNT10B, WNT3, 

immune system process (GO:0002376) 

-LBP, 

localization (GO:0051179) 

-RAB32, TAP2, 

 metabolic process (GO:0008152) 

-SREBF1, BCAR3, CTNNB1, 

multicellular organismal process (GO:0032501) 

-WNT3, WNT10B, 

pigmentation (GO:0043473) 

-RAB32, 

response to stimulus (GO:0050896) 

-LBPLBP, WNT10B, WNT3, 

signaling (GO:0023052) 

-LBP, WNT10B, WNT3.

**Figure 5 biology-11-00555-f005:**
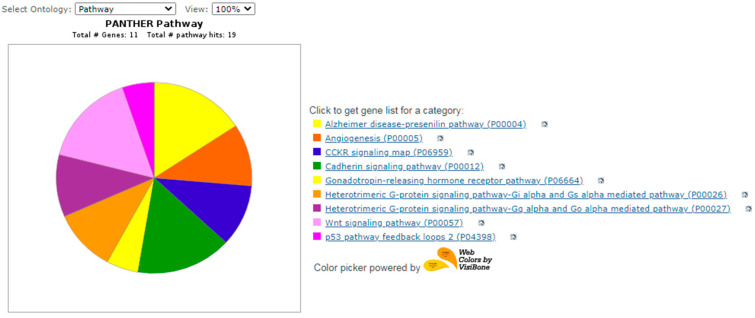
Tested genes and pathway. 

Alzheimer disease-presenilin pathway (P00004) 

-WNT3, CTNNB1, WNT10B, 

Angiogenesis (P00005) 

-WNT10B, CTNNB1, 

CCKR signaling map (P06959) 

-CTNNB1, RGS2, 

Cadherin signaling pathway (P00012) 

-WNT10B, CTNNB1, WNT3, 

Gonadotropin-releasing hormone receptor pathway (P06664) 

-CTNNB1, 

Heterotrimeric G-protein signaling pathway-Gi alpha and Gs alpha mediated pathway (P00026) 

-RGS16, RGS2, 

Heterotrimeric G-protein signaling pathway-Gq alpha and Go alpha mediated pathway (P00027) 

-RGS16, RGS2, 

Integrin signalling pathway (P00034) 

-LAMC1, 

Wnt signaling pathway (P00057) 

-WNT10B,CTNNB1, WNT3, 

p53 pathway feedback loops 2 (P04398) 

-CTNNB1.

**Figure 6 biology-11-00555-f006:**
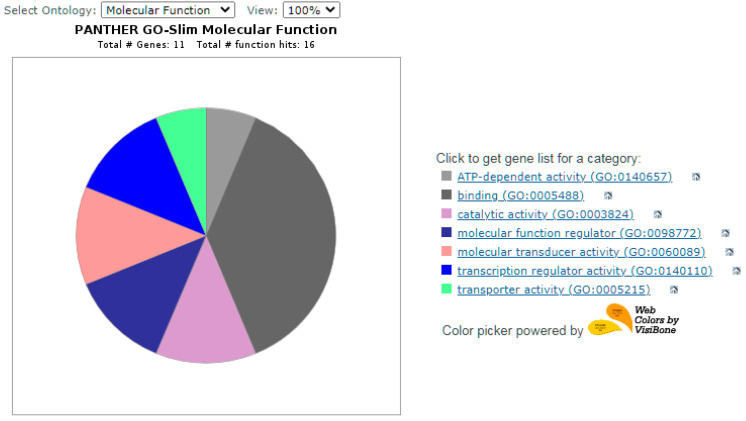
Tested genes and molecular functions. 

ATP-dependent activity (GO:0140657) 

-TAP2, 

binding (GO:0005488) 

-LBP, CHN2, WNT10, BSREBF1, PKP,CTNNB1, 3WNT3, 

catalytic activity (GO:0003824) 

-RAB32, CHN2, TAP2, 

molecular function regulator (GO:0098772) 

-WNT3, CHN2, WNT10B, 

molecular transducer activity (GO:0060089) 

-WNT10B, WNT3, 

transcription regulator activity (GO:0140110) 

-SREBF1, CTNNB1, 

transporter activity (GO:0005215) 

-TAP2.

**Table 1 biology-11-00555-t001:** The list of examined genes.

*WNT3*	WNT3 wingless-type MMTV integration site family, member 3
*RAB32*	RAB32 2, a member of the RAS oncogene family
*WNT10B*	WNT10B wingless-type MMTV integration site family, member 10B
*BCAR3*	BCAR3 breast cancer anti-estrogen resistance 3
*CTNNB1*	CTNNB1 catenin (cadherin-associated protein), beta 1, 88kDa
*CHN2*	CHN2 chimerin (chimaerin) 2
*LAD1*	LAD1 ladinin 1
*RGS16*	RGS16 regulator of G-protein signaling 16
*RGS2*	RGS2 regulator of G-protein signaling 2, 24kDa
*LAMC1*	LAMC1 laminin, gamma 1 (formerly LAMB2)
*LBP*	LBP lipopolysaccharide binding protein
*SR-BP1*	sterol response element-binding protein-1
*TAP2*	Transporter associated with the transformation of antigens
*PKP3*	PKP3 plakophilin 3
*CLDN5*	CLDN5 claudin 5 (transmembrane protein deleted in velocardiofacial syndrome)

**Table 2 biology-11-00555-t002:** Analysis of the correlation of gene expression with the Spearman’s rank-order test. All values are statistically significant (*p* < 0.05).

Gene Names	Statistical Significance	Correlation
WNT3 Pacli 300–WNT10 Pacli 300	0.0338	0.3548
RAB 32 Control–RGS2 Control	0.0027	0.347
RAB 32 Pacli 60–PKP3 Pacli 60	0.0027	−0.4853
PKP Pacli 60–RGS16 Pacli 60	0.0006	0.5464
TREM1 Control–PKP3 Control	0.0311	−0.3598
BCAR3 Pacli 60–PKP3 Pacli 60	0.0036	−0.4726
TREM1 Pacli 60–WNT3 Pacli 60	0.04	0.3439
TREM1Pacli 300–PKP3 Pacli 60	0.047	−0.3334

## Data Availability

The datasets used and/or analyzed during the current study are available from the corresponding author on reasonable request.

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
