# Peer review of "The Expression of Signaling Genes in Breast Cancer Cells"

_biology, 2022, doi:10.3390/biology11040555_

Round 1

Reviewer 1 Report

The authors have addressed my concerns adequately. 

Author Response

The authors have addressed my concerns adequately

Thank you very much for your favorable review.

Reviewer 2 Report

Authors demonstrated the differrent effect of paclitaxel dose using cell line of primary breast cancer. signaling genes were evaluated with cDNA and the result show low dose paclitaxel cause cytostatic effect and high dose paclitaxel cause high cytotoxicity.

English is well written and contents were properly described.

Author Response

Authors demonstrated the differrent effect of paclitaxel dose using cell line of primary breast cancer. signaling genes were evaluated with cDNA and the result show low dose paclitaxel cause cytostatic effect and high dose paclitaxel cause high cytotoxicity.

English is well written and contents were properly described.

Thank you very much for your favorable review.

Reviewer 3 Report

The article 'The expression of signaling genes in breast cancer cells' by Rzymowska et al. proposes investigating the global changes in gene expression using microarray and RT-qPCR after paclitaxel treatment in breast cancer cells.  The area of research is attractive and warrants investigation; however, similar studies have already been published before using a similar strategy in vitro (Gangapuram et al., Int J Mol Sci, 2021; Jurj et al., Cell Physiol Biochem., 2020).  Although the authors have some interesting preliminary data on bona fine and relatively new gene sets, no functional analysis is done to support their role in drug response.  The manuscript does not lay out a reasonable rationale behind the study, and the implications of the results are vague.  The manuscript title and overall study design is not well thought after, and the data is insufficient.  The entire article is written in an incoherent manner such that no connection is established among introduction, results, and discussion.  The discussion is speculative, and the majority of this section is a literature review as functional analyses on candidate genes are not carried out.

The relationship between gene set expressions and clinicopathological parameters in patient samples such as age, gender, histological differentiation, stages, among others, is not established.  No data analysis platform is discussed that was used in gene expression analysis.  Further, no GO/Pathway analysis is performed/discussed to reasonably argue the selection of the gene sets.  No histopathology or cytology data on patient samples are provided either.

Additionally, sentences are not crafted carefully at multiple places throughout the manuscript.  They are worded rather casually and full of grammatical errors.  It becomes difficult for the reader to interpret the outcome.  The writing style also lacks tense verb consistency.

Therefore, this manuscript is premature for publication in 'Biology' at this stage, in my opinion.  Hence, I do not endorse the publication

Author Response

1-The relationship between gene set expressions and clinicopathological parameters in patient samples such as age, gender, histological differentiation, stages, among others, is not established .

AD1. As stated in the text, all patients had early-stage breast cancer and all cancer samples were triple negative and  histopathologically it was NST (non-special type) - that is, invasive ductal carcinoma, the most common type of invasive breast cancer (75-80%) without any special differentiating features that characterize the other categories of breast cancers

Also, all patients were perimenopausal or postmenopausal (median age 56 years) and therefore female.

We added to the text that all the patients were female and were also postmenopausal

2.No data analysis platform is discussed that was used in gene expression analysis. 

AD2.The data analysis is presented in -section –3.4 Analysis of array gene expression matrix, line 225     

The method ( the Quantity One program, version 4.2.1.)  was also described. This method is one of the standard methods used for gene expression analysis and therefore we do not feel the need to discuss this method

Lines 239-244

The statistical analysis was performed using Statistica 10.0 and Microsoft Excel 2015. 239 We computed the arithmetic average and standard deviations for individual groups of 240 genes. In order to assess the significance of the differences between individual groups of 241 genes, we used the Student's t test. For multiple comparisons, one-way analysis of vari-242 ance (ANOVA) followed by Spearman’s rank post-hoc test was applied. Statistical significance 243 was considered as p < 0.05. The rank correlation takes values in the range [- 1, + 1]

3 Further, no GO/Pathway analysis is performed/discussed to reasonably argue the selection of the gene sets. 

We discussed to reasonably justify the choice of gene set in the introduction.

Line-111-116 Resistance to applied chemotherapy, (including  paclitaxel), apart from active drug removal from the cell, is also caused by disturbances in the expression of apoptotic regulatory proteins, loss of resistance to proapoptotic factors, inhibition of apoptosis and activation of autophagy. The studied genes, signaling pathways and protein receptors participate directly or indirectly in the processes responsible for both proliferation and differentiation as well as cell death

Line -59-65

The family of G protein signaling regulatory proteins (RGS) controls the activity of G proteins by accelerating the GTPase activity of the alpha subunits of proteins. Most likely, RGS proteins control the functioning of cancer cells, including tumor progression [11].

BCAR 3 (Breast Cancer Antiestrogen Resistance-3) is found in fibroblasts and is expressed in breast cancer cells not expressing the estrogen receptor (ER). It is a gene that plays a role in the migration and invasion of cancer cells. BCAR3 mRNA expression is undetectable in normal breast cancer tissue 12].

Line 80-85

Paclitaxel, has a different effect on signaling pathways depending on its concentration in the cell,. Low concentrations of taxol inhibit progression through microtubule dysfunction which cannot polymerize properly. It was also found that low drug concentrations induce p53 and p21WAf independent of the RAF-1 pathway. However, at higher concentrations, cell death occurs as a result of blocking the end stage of mitosis and is dependent on the RAF-1 pathway. Paclitaxel mediated cell death may result from Raf-1 dependent and independent pathways [15].

The role of examinated genes was also added at the biginning  in disscusion

Lines 307-315

Healthy and neoplastic tissues have different mechanisms of regulation not only of the expression of multi-drug resistance genes, e.g. MDR1, MRP1, ABCG2, but also genes whose protein products indirectly regulate the activity of ABC transporters, including from the WNT / β-catenin pathways. The CHN2 B2 gene product - chimerin is responsible for the progression of breast cancer. Cancer cells also overexpress PKP3 which is a WNT antagonist. High levels of PKPs are to be associated with a shorter-specific survivel periods. B-catenin activates genes of the transcriptional complex, which leads to tumor progression. [21].The Wnt signaling pathway plays an important role in carcinogenesis. Disturbances in the functioning of this pathway, such as excessive beta-catenin accumulation, are frequently observed in breast cancer cells………

4.No histopathology or cytology data on patient samples are provided either

AD4 All histopathology or cytology data on patient samples are provided line

Lines 132-139

 The study involved cells obtained from breast tissue of 36 patients diagnosed with early stage I and II breast cancer (T1-2N0M0,T – tumor, N – nodes, M – metastasis) according to the WHO (World Human Organisation),  histopathologically results were no special type (NST) and G1 and G2 ductal carcinoma (G ­– grade) according to the previous histopathological classification. The patients presented with triple-negative breast cancer (oestrogen, progesterone and Her2 receptors were all negative). Patients age ranged from 46 to 76 years, with mean age of 56 year. The patients had not received any chemotherapy or hormonotherapy. They were all perimenopausal or postmenopausal and were women.

5.We cannot agree that we only present the literature and do not discuss our results

Lines 332 -339

In the present study, WNT3 expression was higher in the control cells than in the cells incubated with paclitaxel. In this case, the lowest expression occurred at a dose of 300 ng/mL Both doses of paclitaxel resulted in a reduced Wnt3 expression, but the higher dose caused a more significant decrease. Genes encoding the Wnt10 protein, however, exhibited increased activity under the effect of paclitaxel, at both doses. It is believed that overexpression of the Wnt10 gene encoding the protein plays a key role in carcinogenesis through the β-catenin-associated activation pathway [28,29].

Lines 467-474

This increased tumor sensitivity to both doxorubicin and cisplatin chemotherapy. In addition, earlier reports indicating that the canonical Wnt pathway activation is associated with more rapid development of cancer and poor prognosis, were confirmed  [52,53,54].

In the present study, we demonstrated increased CTNNB1 expression after cell incubation with 60 ng/mL of paclitaxel. We also showed higher expression of most genes encoding signaling proteins in cells treated with paclitaxel. The data analysis revealed increased expression of genes encoding receptor proteins in cell membranes [55].

Line 480-485

Our results show different effects of the two doses of paclitaxel (60 and 300 ng/mL) on the expression of signaling genes in breast cancer cells in in vitro settings. The changes in the genes of the test subjects appeared with two different doses of paclitaxel. This may be due to intracellular interactions, but it was directly or indirectly due to the effects of the two different doses of the drug used. The mechanism of these changes is difficult to define unequivocally.

In the discussion, we wanted to describe the role of the studied genes in cancer, and in particular the influence of chemotherapy on their expression, which is related to our research

6.Additionally, sentences are not crafted carefully at multiple places throughout the manuscript.  They are worded rather casually and full of grammatical errors.  It becomes difficult for the reader to interpret the outcome.  The writing style also lacks tense verb consistency.

AD6. The article was corrected once again by a theoretical linguist and English-Polish translator. The following issues has been reviewed and corrected: grammar, sentence structure, phrasing, style, punctuation and spelling. It was done by English translator

Sincerely,

Jolanta Rzymowska

on behalf of all authors

.

Round 2

Reviewer 3 Report

After careful examination of the authors' response and the changes made in the manuscript, I gather that the revised version of the manuscript has improved significantly and addressed some of the major concerns raised in the previous version of the paper. However, the functional validation of the preliminary data on bona fide and relatively new gene sets is still missing, and the data is insufficient. Merely reporting changes in gene expression by a candidate-based approach does not guarantee the functional implications of genes in the system, especially in a system as heterogeneous as breast cancer.

For example, the authors themselves state that Paclitaxel has a different effect on signaling pathways depending on its concentration in the cell. It may affect apoptosis, autophagy or indirectly affect tumor migration or invasion. The authors don't report any phenotype by manipulating candidate gene sets. Further, no data is provided to support the connection between multi-drug resistance genes and WNT. Without the strong experimental evidence, the discussion is still speculative and summarizes results from other studies in relation to the author's experimental system without providing convincing data. I would be happy to endorse the publication if the authors perform functional validation of the candidate genes and carry out a network analysis to figure out the key nodes.

I, therefore, recommend a major revision of the manuscript. 

Author Response

Responses to Reviewer 3

  1. However, the functional validation of the preliminary data on bona fide and relatively new gene sets is still missing, and the data is insufficient. Merely reporting changes in gene expression by a candidate-based approach does not guarantee the functional implications of genes in the system, especially in a system as heterogeneous as breast cancer.

Ad. 1. We added the functional validation of the tested genes in the form of figures 4, 5 and 6 on pages 13, 14 and 15.

We have also added the parts for the BCAR3 gene on page 16, the RAB32 gene, LAMC1 and SR-BP1 on page 17.

  1. Further, no data is provided to support the connection between multi-drug resistance genes and WNT.

Ad. 2. We added the sentence "Elevated Wnt signaling is responsible for resistance to cancer therapy by main-taining the cancer stem cell population and enhancing DNA damage repair. Up-regulation of the WNT pathway after Paclitaxel administration has been ob-served, which may protect cancer cells from cell cycle arrest and apoptosis [28] on page 16 and reference 28 in references chapter.